# Association Analysis in Young and Middle-Aged Mothers—Relation between Expression of Cardiovascular Disease Associated MicroRNAs and Abnormal Clinical Findings

**DOI:** 10.3390/jpm11010039

**Published:** 2021-01-11

**Authors:** Ilona Hromadnikova, Katerina Kotlabova, Ladislav Krofta

**Affiliations:** 1Department of Molecular Biology and Cell Pathology, Third Faculty of Medicine, Charles University, 110 00 Prague, Czech Republic; katerina.kotlabova@lf3.cuni.cz; 2Institute for the Care of the Mother and Child, Third Faculty of Medicine, Charles University, 147 00 Prague, Czech Republic; ladislav.krofta@upmd.eu

**Keywords:** BMI, cardiovascular risk, central obesity, expression, fetal growth restriction, gestational diabetes mellitus, gestational hypertension, hypertension, hypertension on treatment, infertility treatment, microRNA, mothers, preeclampsia, preterm prelabor rupture of membranes, overweight/obesity, spontaneous preterm birth preterm birth, trombophilic gene mutations, whole peripheral blood

## Abstract

The principal goal of the study was to map common postpartal alterations in gene expression of microRNAs associated with diabetes/cardiovascular/cerebrovascular diseases induced by most frequently occurring pregnancy-related complications (gestational hypertension, preeclampsia, fetal growth restriction, gestational diabetes mellitus, preterm prelabor rupture of membranes, or spontaneous preterm birth). In addition, the association analyses between individual abnormal clinical findings (overweight/obesity, central obesity, hypertension, on blood pressure treatment, history of infertility treatment, actual hormonal contraceptive use, the presence of trombophilic gene mutations, actual smoking status, increased serum levels of total cholesterol, HDL (high density lipoprotein) cholesterol, LDL (low density lipoprotein) cholesterol, triglycerides, lipoprotein A, CRP (C-reactive protein), and uric acid, and increased plasma levels of homocysteine) and microRNA expression levels were performed in mothers with respect/regardless to previous course of gestation. The prior exposure to gestational hypertension, preeclampsia, fetal growth restriction, gestational diabetes mellitus, preterm prelabor rupture of membranes, or spontaneous preterm birth caused that a significant proportion of mothers (52.42% at 90.0% specificity) had substantially altered microRNA expression profile, which might originate lifelong cardiovascular risk. 26 out of 29 tested microRNAs were up-regulated in mothers with a history of such complicated pregnancies. MicroRNA expression profiles were also able to differentiate between mothers with normal and abnormal clinical findings (BMI (body mass index), waist circumference, systolic blood pressure, on blood pressure treatment, history of infertility treatment, and the presence of trombophilic gene mutations) irrespective of previous course of gestation. The treatment of hypertension even intensified upregulation of some microRNAs (miR-24-3p, and miR-342-3p) already present in women after complicated pregnancies. Newly, the presence of overweight/obesity (miR-155-5p), systolic hypertension (miR-92a-3p, and miR-210-3p), treatment for infertility (miR-155-5p), and treatment for hypertension (miR-210-3p) induced upregulation of several microRNAs. In general, mothers after complicated pregnancies are at increased risk of development of cardiovascular complications. Especially those mothers indicated to have postpartally altered microRNA expression profiles might be considered as a highly risky group that would benefit from dispensarization and implementation of primary prevention strategies.

## 1. Introduction

A previous occurrence of pregnancy-related complications is associated with an increased risk of the onset of series of diseases involving hypertension [1,2,3,4,5,6,7,8], diabetes mellitus [1,2,3,4,9,10], metabolic syndrome [11,12], kidney diseases [2], atherosclerosis [13,14], ischemic heart disease [2,5,6,15,16,17,18,19], myocardial infarcts [2,3,4,7,16,17], heart failure [2,3,4,7], stroke [2,3,4,5,6,7,15,16,17,19], and deep venous thrombosis [1,5,6].

Recently, we have shown that a proportion of mothers with a history of pregnancy-related complications such as gestational hypertension (GH), preeclampsia (PE), fetal growth restriction (FGR), and gestational diabetes mellitus (GDM) had alterations in microRNA expression profiles in whole peripheral blood (leukocytes) that may contribute besides other factors to the onset of diabetes mellitus, cardiovascular and cerebrovascular diseases [20,21]. 

A previous occurrence of GH, PE, FGR, and GDM was demonstrated to be associated in a proportion of mothers with alterations in gene expression of miR-1-3p, miR-17-5p, miR-20a-5p, miR-20b-5p, miR-29a-3p, miR-100-5p, miR-125b-5p, miR-126-3p, miR-130b-3p, miR-133a-3p, miR-143-3p, miR-145-5p, miR-146a-5p, miR-181a-5p, miR-199a-5p, miR-221-3p, and miR-499a-5p [20,21].

Furthermore, a history of GDM was shown to be associated with dysregulation of miR-16-5p, miR-21-5p, miR-23a-3p, miR-24-3p, miR-26a-5p, miR-103a-3p, miR-195-5p, miR-342-3p, and miR-574-3p in a substantial proportion of mothers [21].

The principal goal of the present study was to describe common postpartal alterations in gene expression of microRNAs associated with diabetes and cardiovascular/cerebrovascular diseases in whole peripheral venous blood (leukocytes) of mothers induced by most frequently occurring pregnancy-related complications such as GH, PE, FGR, GDM, preterm prelabor rupture of membranes or spontaneous preterm birth.

In addition, we extended the study to perform the association analyses to see the impact of individual abnormal clinical findings (overweight/obesity, central obesity, hypertension, on blood pressure treatment, history of infertility treatment, actual hormonal contraceptive use, the presence of trombophilic gene mutations, actual smoking status, aberrant serum levels of total cholesterol, HDL cholesterol, LDL cholesterol, triglycerides, lipoprotein A, CRP, and uric acid, and increased plasma levels of homocysteine) on microRNA expression profiles in mothers after normal and complicated pregnancies.

Furthermore, we assessed to what extent individual abnormal clinical findings (overweight/obesity, central obesity, hypertension, on blood pressure treatment, history of infertility treatment, actual hormonal contraceptive use, the presence of trombophilic gene mutations, actual smoking status, aberrant serum levels of total cholesterol, HDL cholesterol, LDL cholesterol, triglycerides, lipoprotein A, CRP, and uric acid, and increased plasma levels of homocysteine) might influence already present aberrant microRNA expression profiles in mothers after complicated pregnancies.

A shortlist of 29 microRNAs, which had been reported to play a key role in the inducement and progress of diabetes mellitus and cardiovascular/cerebrovascular diseases, was selected for our previous and currently ongoing studies (Appendix A).

## 2. Results

### 2.1. Substantially Altered Expression Profile of Diabetes/Cardiovascular/Cerebrovascular Disease Associated Micrornas in Mothers after Complicated Pregnancies

Firstly, we compared microRNA gene expression profile between mothers after normal and complicated pregnancies irrespective of the type of pregnancy-related complication (gestational diabetes mellitus, gestational hypertension, preeclampsia, fetal growth restriction, preterm prelabor rupture of membranes, and/or spontaneous preterm birth). Significantly increased expression of 26 out of 29 tested microRNAs (miR-1-3p, miR-16-5p, miR-17-5p, miR-20a-5p, miR-20b-5p, miR-21-5p, miR-23a-3p, miR-24-3p, miR-26a-5p, miR-29a-3p, miR-100-5p, miR-103a-3p, miR-125b-5p, miR-126-3p, miR-130b-3p, miR-133a-3p, miR-143-3p, miR-145-5p, miR-146a-5p, miR-181a-5p, miR-195-5p, miR-199a-5p, miR-221-3p, miR-342-3p, miR-499a-5p, and miR-574-3p) was identified in mothers after complicated pregnancies using the M-W and ROC curve analyses. Aberrant microRNA expression profile differentiated between mothers after normal and complicated pregnancies with a sensitivity ranging from 5.09% to 29.26% at 10.0% false positive rate (FPR) (Appendix A).

Screening based on a combination of 24/26 microRNAs (without miR-24-3p and miR-342-3p) showed that at 10.0% FPR 52.42% mothers after complicated pregnancies had substantially altered postpartal microRNA expression profile, which may indicate an increased risk of later development of diabetes and/or cardiovascular/cerebrovascular diseases (AUC 0.798, *p* < 0.001, sensitivity 75.57%, specificity 69.66%, cut off >0.758136883) (Figure 1).

Subsequently, we compared microRNA expression profile between individual groups of mothers with respect to actual clinical findings (BMI, waist circumference values, systolic and diastolic blood pressure values, on blood pressure treatment, history of infertility treatment, actual hormonal contraceptive use, the presence of trombophilic mutations, actual smoking status, total serum cholesterol levels, serum HDL cholesterol levels, serum LDL cholesterol levels, serum triglycerides levels, serum lipoprotein A levels, serum CRP levels, plasma homocysteine levels, and serum uric acid levels) regardless of and with respect to the course of previous gestation (normal course of gestation vs. gestation complicated with gestational diabetes mellitus, gestational hypertension, preeclampsia, fetal growth restriction, preterm prelabor rupture of membranes, and/or spontaneous preterm birth). Just the results that reached a statistical significance or displayed a trend towards statistical significance are presented below.

### 2.2. No Association between Postpartal Expression of Diabetes/Cardiovascular/Cerebrovascular Disease Associated Micrornas and Actual Hormonal Contraceptive Use, Active Smoking of Cigarettes, Total Serum Cholesterol Levels, Serum HDL Cholesterol Levels, Serum LDL Cholesterol Levels, Serum Triglycerides Levels, Serum Lipoprotein A levels, Serum CRP Levels, Serum Uric Acid Levels, and Plasma Homocysteine Levels

No association between expression of diabetes/cardiovascular/cerebrovascular disease associated microRNAs and actual hormonal contraceptive use, and active smoking of cigarettes was observed in young and middle-aged mothers. In detail, there was no difference in microRNA gene expression between never users and current users of any hormonal contraceptives (combined oral contraceptives, the progestogen-only pills, oral selective estrogen receptor modulators, intrauterine device, contraceptive ring, or the contraceptive implant). Furthermore, there was no difference in microRNA gene expression between never smokers and current smokers of tobacco cigarettes.

In addition, no association between microRNA expression levels and serum levels of total cholesterol, HDL cholesterol, LDL cholesterol, triglycerides, lipoprotein A, CRP, and uric acid was found. MicroRNA gene expression did not also differ in relation to plasma homocysteine levels. In detail, there was no difference in microRNA gene expression between mothers with abnormal and normal serum or plasma levels of these biochemical analytes.

### 2.3. Postpartal Expression Profile of miR-155-5p Differentiates between Overweight/Obese Mothers and Mothers with Normal BMI Values

Overall, 183 out of 482 (37.97%) tested mothers were confirmed to be overweight/obese. In a group of mothers after complicated pregnancies, the incidence of overweight/obesity reached 40.97% (161 out of 393 tested mothers).

MiR-155-5p showed a trend towards increased expression in overweight/obese mothers when the comparison to mothers with normal BMI values regardless of a history of gestation was performed (*p* = 0.074). In a group of mothers after complicated pregnancies only, even higher miR-155-5p expression rates were detected in overweight/obese individuals (*p* = 0.037). The performance of ROC curve analyses revealed that the expression levels of miR-155-5p were increased in 18.03% overweight/obese mothers irrespective of the course of previous pregnancies and in 19.88% overweight/obese mothers after complicated pregnancies (Appendix A).

### 2.4. Postpartal Expression Profile of miR-195-5p Differentiates between Mothers with Central Obesity and Mothers with Normal Values of Waist Circumference

Overall, 213 out of 482 (44.19%) tested mothers were confirmed to have abnormal values of waist circumference.

Women with central obesity showed increased expression levels of miR-195-5p (*p* = 0.049), nevertheless only irrespective of the course of previous gestation. The performance of ROC curve analyses revealed that at 10.0% FPR 15.49% mothers with central obesity had substantially altered expression profile of miR-195-5p (Appendix A).

### 2.5. Postpartal Expression Profile of miR-17-5p, miR-24-3p, miR-92a-3p, miR-126-3p, miR-130b-3p, miR-181a-5p, and miR-210-3p Differentiates between Mothers with Systolic Hypertension and Mothers with Normal SBP Values

Overall, 28 out of 482 (5.81%) tested mothers were confirmed to have systolic hypertension. 27 out of 28 women with identified systolic hypertension had a history of complicated pregnancy.

7 out of 29 tested microRNAs (miR-17-5p, miR-24-3p, miR-92a-3p, miR-126-3p, miR-130b-3p, miR-181a-5p, and miR-210-3p) showed increased expression or a trend towards increased expression in mothers with systolic hypertension when the comparison to mothers with normal SBP values regardless of a history of gestation was performed. The performance of ROC curve analyses revealed that these individual microRNAs were able to differentiate at 10.0% FPR between mothers with normal and abnormal SBP values with a sensitivity ranging from 7.14% to 14.29% (Appendix A).

Screening based on the combination of miR-17-5p and miR-130b-3p showed the best accuracy from all microRNA combinations. This combined microRNA screening showed that at 10.0% FPR 14.29% mothers with systolic hypertension had substantially altered expression profile of miR-17-5p and miR-130b-3p (AUC 0.631, *p* = 0.004, sensitivity 82.14%, specificity 46.04%, cut off >0.05337161) (Figure 2).

The accuracy of screening based on the combination of miR-17-5p and miR-130b-3p was nearly equal to the accuracy of miR-130b-3p, the individual microRNA with the best performance (AUC 0.623, *p* = 0.009, sensitivity 14.29% at 10.0% FPR) (Appendix A).

### 2.6. Postpartal Expression Profile of miR-24-3p, miR-210-3p, miR-221-3p, and miR-342-3p Differentiates between Mothers on Blood Pressure Treatment and Mothers without Medication

Overall, 15 out of 482 (3.11%) tested mothers were on blood pressure treatment. In a group of mothers after complicated pregnancies, medication to lower blood pressure was given to 14 out of 393 tested mothers (3.56%).

In a mixed group of mothers after normal and complicated pregnancies, the expression of microRNAs differed significantly or was on a border of statistical significance (miR-24-3p, miR-210-3p, miR-221-3p, and miR-342-3p) between individuals on blood pressure treatment and without medication. The performance of ROC curve analyses revealed that these individual microRNAs were able to differentiate at 10.0% FPR between mothers with and without medication with a sensitivity ranging from 26.67% to 33.33% (Appendix A).

Irrespective of a history of gestation, screening based on individual microRNAs (miR-210-3p or miR-221-3p) showed nearly the equal performance (miR-210-3p: AUC 0.652, *p* = 0.077, sensitivity 26.67% at 10.0% FPR; miR-221-3p: AUC 0.649, *p* = 0.071, sensitivity 33.33% at 10.0% FPR) as the combination of microRNAs with the best performance from various microRNA combinations (miR-210-3p and miR-221-3p). Screening based on combination of miR-210-3p and miR-221-3p revealed that at 10.0% FPR 33.33% mothers had substantially altered microRNA expression profile (AUC 0.649, *p* = 0.073, sensitivity 33.33%, specificity 98.50%, cut off >0.075017814) (Figure 3).

In a group of mothers after complicated pregnancies only, even higher expression rates of miR-24-3p, miR-210-3p, and miR-342-3p were detected in individuals on blood pressure treatment. The performance of ROC curve analyses revealed that the expression levels of miR-24-3p (28.57%), miR-210-3p (28.57%), and miR-342-3p (28.57%) were increased in a proportion of mothers after complicated pregnancies, which required blood pressure treatment (Appendix A).

In a group of mothers after complicated pregnancies only, screening based on one individual microRNA (miR-210-3p: AUC 0.693, *p* = 0.019, 28.57% at 10.0% FPR) was superior over the performance of various combinations of individual microRNAs. The screening based on combination of miR-24-3p and miR-210-3p, which showed the best performance from various microRNA combinations, also revealed that at 10.0% FPR 28.57% mothers on blood pressure treatment had substantially altered postnatal expression profile of these microRNAs (AUC 0.675, *p* = 0.030, sensitivity 85.71%, specificity 49.34%, cut off >0.027905823), but the area under the ROC curve was slightly lower (Appendix A).

### 2.7. Postpartal Expression Profile of miR-155-5p Differentiates between Mothers with Respect to Infertility Treatment

Overall, 70 out of 482 (14.52%) tested mothers have been treated for infertility. In a group of mothers after complicated pregnancies, infertility treatment program was passed by 66 out of 393 tested mothers (16.79%).

In a mixed group of mothers after normal and complicated pregnancies, the expression of miR-155-5p showed a trend towards increased expression in mothers with pregnancies after infertility treatment when compared to mothers after spontaneous conception (*p* = 0.099). In a group of mothers after complicated pregnancies only, even higher miR-155-5p expression rates were detected in a group of mothers with pregnancies after infertility treatment (*p* = 0.041). The performance of ROC curve analyses revealed that at 10.0% FPR the expression levels of miR-155-5p were increased in 10.0% mothers with pregnancies after infertility treatment irrespective of the course of previous pregnancies and in 10.61% mothers after complicated pregnancies, who passed infertility treatment program (Appendix A).

### 2.8. Postpartal Expression Profile of miR-1-3p, miR-16-5p, miR-20b-5p, miR-24-3p, miR-26a-5p, miR-103a-3p, miR-125b-5p, miR-130b-3p, miR-143-3p, miR-146a-5p, miR-199a-5p, miR-221-3p, and miR-342-3p Differentiates between Mothers with and without the Presence of Trombophilic Gene Mutations

Overall, 37 out of 482 (7.68%) tested mothers were identified to be carriers of trombophilic gene mutations, in which case all of these 37 carriers of trombophilic gene mutations had complicated pregnancies.

Women with the presence of trombophilic gene mutations showed increased expression or a trend towards increased expression levels of miR-1-3p (*p* = 0.062), miR-16-5p (*p* = 0.039), miR-20b-5p (*p* = 0.077), miR-24-3p (*p* = 0.061), miR-26a-5p (*p* = 0.056), miR-103a-3p (*p* = 0.027), miR-125b-5p (*p* = 0.070), miR-130b-3p (*p* = 0.054), miR-143-3p (*p* = 0.009), miR-146a-5p (*p* = 0.063), miR-199a-5p (*p* = 0.049), miR-221-3p (*p* = 0.053), and miR-342-3p (*p* = 0.056). The performance of ROC curve analyses revealed that at 10.0% FPR a significant proportion of mothers with the presence of trombophilic gene mutations had substantially altered expression profile of miR-1-3p (21.62%), miR-16-5p (24.32%), miR-20b-5p (21.62%), miR-24-3p (24.32%), miR-26a-5p (24.32%), miR-103a-3p (18.92%), miR-125b-5p (18.92%), miR-130b-3p (21.62%), miR-143-3p (18.92%), miR-146a-5p (18.92%), miR-199a-5p (24.32%), miR-221-3p (24.32%), and miR-342-3p (21.62%) (Appendix A).

Screening based on one individual microRNA (miR-143-3p: AUC 0.628, *p* = 0.007, 18.92% at 10.0% FPR) (Appendix A) was superior over the performance of various combinations of individual microRNAs. The screening based on combination of miR-16-5p, miR-103a-3p, and miR-143-3p, which showed the best performance from various microRNA combinations, also revealed that at 10.0% FPR 18.92% mothers, who are carriers of trombophilic gene mutations, had substantially altered postpartal expression profile of these microRNAs (AUC 0.617, *p* = 0.016, sensitivity 54.05%, specificity 70.11%, cut off >0.07293491), but the area under the ROC curve was slightly lower (Appendix A).

### 2.9. The Effect of Physical Activity on Postpartal microRNA Expression Profile, BMI Values, SBP and DBP Values

Patients were divided into two groups, those ones who were not used to run any sport activity (*n* = 295) and those ones who were running some sport activities (*n* = 183). Minimum women were running sport activities on daily basis or several times per week. Most women, who did any sport activity, made it mainly from time to time for leisure.

The association analysis between physical activity and microRNA gene expression showed decreased levels of microRNA gene expression in women used to run any sport activity from time to time, however any analysis did not achieve statistical significance (Appendix A).

The association analysis between physical activity and BMI values also showed decreased levels of BMI values in women used to run any sport activity from time to time, however the analysis has not yet achieved the statistical significance (Mann-Whitney test, *p* = 0.186) (Appendix A). Similarly, the impact of physical activity on BMI values (below 25 vs. above 25) has not yet achieved statistical significance, when logistic regression was used (OR 0.774, 95% CI: 0.5280 to 1.1359, *p* = 0.1892).

On the other hand, physical activity mentioned above had no impact on SBP values (Mann-Whitney test, *p* = 0.852) (Appendix A) in our cohort of patients. Similarly, physical activity mentioned above had also no impact on DBP values (Mann-Whitney test, *p* = 0.735) (Appendix A). Likewise, no effect of occasional physical activity was observed in women with the presence of systolic hypertension (above 140 mmHg vs. below 140 mmHg) (OR 0.8908, 95% CI: 0.4019 to 1.9744, *p* = 0.775) and diastolic hypertension (above 90 mmHg vs. below 90 mmHg) (OR 0.8527, 95% CI: 0.4425 to 1.6432, *p* = 0.632).

## 3. Discussion

Firstly, postpartal gene expression profiles of 29 selected microRNAs known to be associated with diabetes mellitus and cardiovascular/cerebrovascular diseases [Appendix A] were compared between mothers after normal and complicated pregnancies. The group of complicated pregnancies consisted of women with a history of most frequently occurring pregnancy-related complications such as gestational diabetes mellitus, gestational hypertension, preeclampsia, fetal growth restriction, preterm prelabor rupture of membranes, and spontaneous preterm birth.

Subsequently, we made comparison of microRNA expression profiles between individual groups of mothers with respect to actual clinical findings and anamnesis data regardless of previous occurrence of pregnancy-related complications. The association analyses to evaluate the impact of individual abnormal clinical findings (overweight/obesity, central obesity, hypertension, on blood pressure treatment, history of infertility treatment, actual hormonal contraceptive use, the presence of trombophilic gene mutations, actual smoking status, aberrant serum levels of total cholesterol, HDL cholesterol, LDL cholesterol, triglycerides, lipoprotein A, CRP, and uric acid, and increased plasma levels of homocysteine) on microRNA expression profiles in mothers irrespective the course of previous gestation were performed.

We also compared microRNA gene expression profiles between equal groups of mothers (mothers after normal pregnancies only and mothers after complicated pregnancies only) with respect to actual clinical findings and anamnesis data. Nevertheless, in case of mothers after normal pregnancies the comparison of microRNA gene expression profiles might be performed due to low number of positive cases only between the individuals with respect to the occurrence of overweight/obesity and central obesity. No difference in microRNA expression levels between the overweight/obese women and women with normal BMI values in a group of women with a history of normal gestation was observed. Similarly, no difference in microRNA expression levels between women with central obesity and women with normal waist circumference values in a group of women with a history of normal gestation was identified.

Overall, significantly altered microRNA expression profile (26 out of 29 tested microRNAs) was identified in a substantial proportion of mothers (52.42% at 10.0% FPR) after complicated pregnancies when compared with mothers after normal pregnancies. All microRNAs (miR-1-3p, miR-16-5p, miR-17-5p, miR-20a-5p, miR-20b-5p, miR-21-5p, miR-23a-3p, miR-24-3p, miR-26a-5p, miR-29a-3p, miR-100-5p, miR-103a-3p, miR-125b-5p, miR-126-3p, miR-130b-3p, miR-133a-3p, miR-143-3p, miR-145-5p, miR-146a-5p, miR-181a-5p, miR-195-5p, miR-199a-5p, miR-221-3p, miR-342-3p, miR-499a-5p, and miR-574-3p) were upregulated in whole peripheral blood (leukocytes) of mothers with a history of pregnancy-related complications. We propose that complicated pregnancy induced alterations in expression of microRNAs playing a role in the pathogenesis of diabetes mellitus and diverse cardiovascular/cerebrovascular diseases might originate a solid base for predisposition to later development of hypertension, diabetes mellitus, metabolic syndrome, kidney diseases, atherosclerosis, ischemic heart disease, myocardial infarcts, heart failure, stroke, and deep venous thrombosis, since increased incidence of these disorders has already been reported to be associated with previous occurrence of pregnancy-related complications [1,2,3,4,5,6,7,8,9,10,11,12,13,14,15,16,17,18,19,20,21].

For that reason we consider mothers with a history of any type of most frequently occurring pregnancy-related complications indicated to have postpartally altered microRNA expression profile in their whole peripheral blood (leukocytes) as a highly risky group that might benefit from dispensarization and implementation of primary prevention strategies.

Concerning the results of association analyses, we could see that the presence of overweight/obesity, central obesity, trombophilic gene mutations, systolic hypertension, the treatment for hypertension, and the treatment for infertility had also certain influence on microRNA expression profiles, however this impact was of diverse extent depending on the type of clinical finding. Up-regulation of miR-155-5p was observed in a part of overweight/obese women irrespective of the course of previous gestation (18.03% at 10.0% FPR) and in a part of overweight/obese women after complicated pregnancies (19.88% at 10.0% FPR). In addition, increased expression of miR-195-5p was found in a proportion of women with waist circumference values over 80 cm, but only in a mixed group of women regardless of a history of gestation (15.49% at 10.0% FPR). Moreover, association analysis pointed out to the fact that alteration in gene expression of miR-155-5p might be also induced by infertility treatment in a proportion of women (10.0% women irrespective of the course of previous gestation at 10.0% FPR, and 10.61% women after complicated pregnancies only at 10.0% FPR).

Parallel, altered expression of miR-17-5p (7.14% at 10.0% FPR), miR-24-3p (10.71% at 10.0% FPR), miR-92a-3p (14.29% at 10.0% FPR), miR-126-3p (14.29% at 10.0% FPR), miR-130b-3p (14.29% at 10.0% FPR), miR-181a-5p (10.71% at 10.0% FPR), and miR-210-3p (14.29% at 10.0% FPR) was detected in a proportion of women with systolic hypertension, but again only in a mixed group of women after normal and complicated pregnancies. Nevertheless, a larger proportion of women on blood pressure treatment showed aberrant expression of miR-24-3p (26.67% at 10.0% FPR), miR-210-3p (26.67% at 10.0% FPR), miR-221-3p (33.33% at 10.0% FPR), and miR-342-3p (26.67% at 10.0% FPR). A history of complicated pregnancies had even higher impact on microRNA expression levels in whole peripheral blood of mothers. The treatment for hypertension even intensified upregulation of miR-24-3p (28.57% at 10.0% FPR), and miR-342-3p (28.57% at 10.0% FPR) in a group of women with a history of complicated pregnancies. Newly, the treatment for hypertension induced upregulation of miR-210-3p (26.67% women irrespective of the course of previous gestation at 10.0% FPR, and 28.57% women after complicated pregnancies only at 10.0% FPR).

Interestingly, microRNA expression profiles were also able to differentiate between women with the presence and the absence of trombophilic gene mutations, but only in a total group of women (a mixture of women after normal and complicated pregnancies). A certain proportion of women with the presence of trombophilic gene mutations showed upregulation of multiple microRNAs at 10.0% FPR miR-1-3p (21.62%), miR-16-5p (24.32%), miR-20b-5p (21.62%), miR-24-3p (24.32%), miR-26a-5p (24.32%), miR-103-3p (18.92%), miR-125b-5p (18.92%), miR-130b-3p (21.62%), miR-143-3p (18.92%), miR-146a-5p (18.92%), miR-199a-5p (24.32%), miR-221-3p (24.32%), and miR-342-3p (21.62%).

No association between expression of diabetes/cardiovascular/cerebrovascular disease associated microRNAs and actual hormonal contraceptive use, and active smoking of cigarettes was observed in young and middle-aged mothers. In addition, no association between microRNA expression levels and serum levels of total cholesterol, HDL cholesterol, LDL cholesterol, triglycerides, lipoprotein A, CRP, and uric acid was found. MicroRNA gene expression did not also differ in relation to plasma homocysteine levels.

From the data resulting from this study, it is apparent, that the evaluation of potential cardiovascular risk based on the presence of aberrant microRNA expression profiles in peripheral blood leukocytes represents a feasible non-invasive approach that might be implemented into the routine postpartal care of young and middle-aged women with a history of complicated pregnancies.

Some therapeutic approaches (for example the treatment for hypertension) might also contribute to modifications of postpartal microRNA expression profiles in peripheral blood leukocytes of women. Intensification of upregulation of several microRNAs (miR-24-3p, and miR-342-3p) already present in a group of women after complicated pregnancies was observed in case of women that had been treated for hypertension. On the other hand, therapeutic approaches (for example the treatment for hypertension or the treatment for infertility) and some clinical findings (overweight/obesity, and systolic hypertension) might induce novel alterations in microRNA gene expression profiles which had not been observed to be associated with the previous occurrence of pregnancy-related complications. That is the case of miR-155-5p that was observed to be upregulated in overweight/obese women and in women with a history of infertility treatment. That is also the case of miR-92a-3p and miR-210-3p that were observed to be upregulated in women with systolic hypertension, and the case of miR-210-3p, which was upregulated in women on blood pressure treatment.

## 4. Materials and Methods

### 4.1. Participants

The prospective study running from August 2016–October 2020 included Caucasian women after normal pregnancies (*n* = 89) and pregnancies complicated with gestational diabetes mellitus (*n* = 112), gestational hypertension (*n* = 47), preeclampsia (*n* = 123), fetal growth restriction (*n* = 38), preterm prelabor rupture of membranes (*n* = 39), and spontaneous preterm birth (*n* = 34). The clinical data of women are displayed in Table 1.

The inclusion and exclusion criteria involving clinical definitions of pregnancy-related complications are specified in detail in previously published papers [22,23,24,25,26,27,28,29] and also in Appendix A of this publication.

Informed consent was signed by all participants. The following Ethics Committees (the Institute for the Care of the Mother and Child, and the Third Faculty of Medicine, Charles University) approved the study (grant no. AZV 16-27761A, Long-term monitoring of complex cardiovascular profile in the mother, fetus and offspring descending from pregnancy-related complications, dates of approval: 27.3.2014 and 28.5.2015). All procedures were in compliance with the Helsinki Declaration of 1975, as revised in 2000.

### 4.2. Blood Pressure, BMI and Waist Circumference Measurements

Detailed information on standardized BP, BMI and waist circumference measurements are provided in Appendix A and in our previous publications [30,31].

### 4.3. Biological Sampling and Data Collection

For detailed information on biological sampling and testing please see Appendix A and our previous publications [30,31].

Information on a history of infertility treatment, the presence of trombophilic gene mutations, actual hormonal contraceptive use, and actual smoking status were acquired from patients’ records deposited in hospital information system and during the time of the patients’ visit.

### 4.4. Processing of Samples, Reverse Transcription, and Relative Quantification of microRNAS

For detailed information on processing of samples, reverse transcription, and relative quantification of microRNAs please see Appendix A and previous publications [20,21,32,33].

### 4.5. Statistical Analysis

For detailed information on statistical analysis please see Appendix A and previous publications [20,21,34].

## 5. Conclusions

In conclusion, microRNA expression profiles characteristic for patients with diabetes mellitus, cardiovascular/cerebrovascular diseases are also present in whole peripheral venous blood (white blood cells) of mothers after complicated pregnancies (gestational diabetes mellitus, gestational hypertension, preeclampsia, fetal growth restriction, preterm prelabor rupture of membranes, and/or spontaneous preterm birth). This finding indicates that previous occurrence of pregnancy-related complications regardless of its type may predispose mothers to later development of diabetes mellitus, and cardiovascular/cerebrovascular diseases. The treatment for hypertension has an additional emphasizing impact on already altered microRNA expression profile in a group of mothers after complicated pregnancies. MicroRNA expression profiles are also able to differentiate between mothers with normal and abnormal clinical findings (BMI, waist circumference, systolic blood pressure, on blood pressure treatment, history of infertility treatment, and the presence of trombophilic gene mutations) irrespective of previous course of gestation. The presence of overweight/obesity (miR-155-5p), systolic hypertension (miR-92a-3p and miR-210-3p), treatment for infertility (miR-155-5p), and treatment for hypertension (miR-210-3p) induces alterations in gene expression of several microRNAs. Consecutive large-scale studies are needed to verify the findings resulting from this pilot study.

## 6. Patents

National patent granted—Industrial Property Office, Czech Republic (Patent n. 308178).

International patent filed—Industrial Property Office, Czech Republic (PCT/CZ2019/050051).

## Figures and Tables

**Figure 1 jpm-11-00039-f001:**
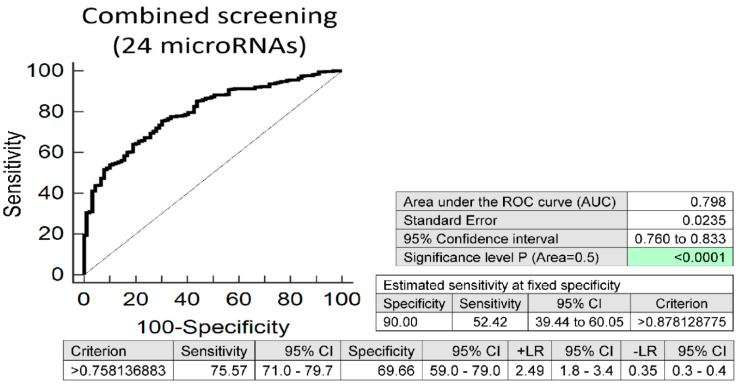
Aberrant microRNA expression profile in mothers after complicated pregnancies (gestational diabetes mellitus, gestational hypertension, preeclampsia, fetal growth restriction, preterm prelabor rupture of membranes, and/or spontaneous preterm birth). Combined microRNA screening revealed that at 10.0% FPR 52.42% mothers with a history of any pregnancy-related complication had substantially altered expression profile of miR-1-3p, miR-16-5p, miR-17-5p, miR-20a-5p, miR-20b-5p, miR-21-5p, miR-23a-3p, miR-26a-5p, miR-29a-3p, miR-100-5p, miR-103a-3p, miR-125b-5p, miR-126-3p, miR-130b-3p, miR-133a-3p, miR-143-3p, miR-145-5p, miR-146a-5p, miR-181a-5p, miR-195-5p, miR-199a-5p, miR-221-3p, miR-499a-5p, and miR-574-3p.

**Figure 2 jpm-11-00039-f002:**
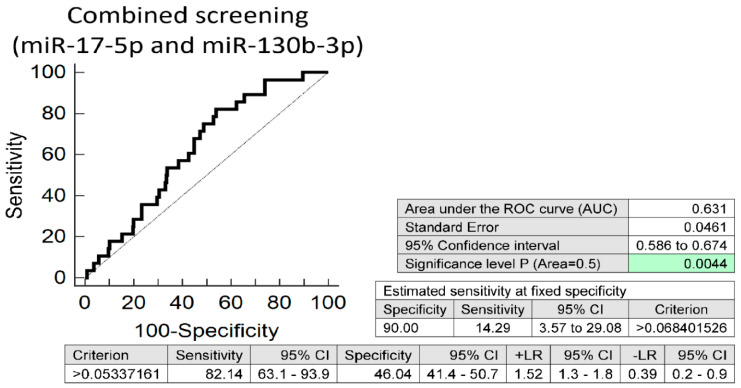
Aberrant microRNA expression profile in mothers with systolic hypertension. Irrespective of the course of previous gestation (normal and complicated pregnancies altogether), screening based on combination of miR-17-5p and miR-130b-3p showed the best performance from various microRNA combinations. At 10.0% FPR 14.29% mothers with systolic hypertension had substantially altered expression profile of miR-17-5p and miR-130b-3p.

**Figure 3 jpm-11-00039-f003:**
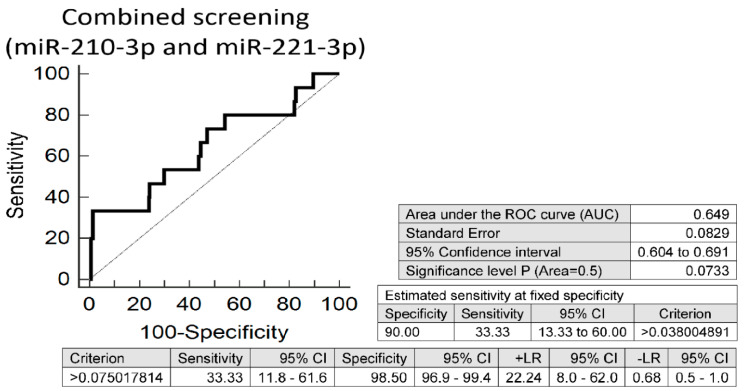
Aberrant microRNA expression profile in mothers on blood pressure treatment. Irrespective of the course of gestation (normal and complicated pregnancies altogether), screening based on combination of miR-210-3p and miR-221-3p showed the best performance from various microRNA combinations. At 10.0% FPR 33.33% mothers on blood pressure treatment had substantially altered expression profile of miR-210-3p and miR-221-3p.

**Table 1 jpm-11-00039-t001:** Characteristics of cases and controls.

	NP (*n* = 89)	PE (*n* = 123)	FGR (*n* = 38)	GH (*n* = 47)	GDM (*n* = 112)	PTB (*n* = 34)	PPROM (*n* = 39)	*p* ^1^	*p* ^2^	*p* ^3^	*p* ^4^	*p* ^5^	*p* ^6^
**At Follow-Up**													
Age (years)	38 (29–50)	38 (28–52)	38 (26–45)	38 (31–58)	39 (31–50)	37 (25–43)	37 (26–47)	1.000	1.000	1.000	1.000	1.000	1.000
Time since index pregnancy (years)	5 (3–11)	4 (3–11)	4 (3–10)	4 (3–10)	5 (4–10)	5 (3–8)	5 (3–9)	1.000	0.274	0.318	1.000	1.000	1.000
Height (cm)	167.0 (153–181)	166.0 (152–183)	166.5 (156–173)	167.5 (157.5–180)	166.0 (151–178.5)	168.5 (155–182)	164.0 (146–176.5)	1.000	1.000	1.000	1.000	1.000	1.000
Weight (kg)	62.7 (46–109)	70.0 (41–121)	62.5 (46.7–107)	72.3 (47–128)	64.5 (43.4–98.7)	61.7 (49.9–87.4)	68.8 (43.5–103.4)	**0.004**	1.000	**0.002**	1.000	1.000	1.000
BMI (kg/m^2^)	22.23 (17.7–39.08)	24.31 (16.22–40.9)	22.25 (17.06–36.76)	25.92 (17.91–46.45)	23.51 (17.39–34.37)	21.09 (18.25–29.9)	24.49 (18.25–34.35)	**<0.001**	1.000	**<0.001**	0.233	1.000	0.344
Waist circumference (cm)	74 (63–117)	79 (64–120)	75 (63–115)	84 (65–131)	80 (63–105)	74 (65–98)	77 (63–105)	**<0.001**	1.000	**<0.001**	**0.008**	1.000	1.000
Systolic BP (mmHg)	112 (87–138)	123 (95–161)	117 (96–163)	126 (109–166)	112.5 (92–158)	110.5 (98–144)	113 (99–189)	**<0.001**	0.246	**<0.001**	1.000	1.000	1.000
Diastolic BP (mmHg)	71 (55–91)	77 (54–105)	77 (59–110)	80 (61–104)	75 (55–109)	72 (61–83)	74 (64–112)	**<0.001**	**0.098**	**<0.001**	0.877	1.000	1.000
Heart rate at rest (n/min)	70 (44–107)	73 (56–92)	73 (57–94)	74 (51–115)	74 (53–102)	74.5 (59–90)	74 (59–94)	1.000	1.000	1.000	0.311	0.828	1.000
Serum total cholesterol (mmol/L)	5.04 (3.49–7.22)	5.07 (3.33–7.33)	4.92 (3.59–7.54)	5.04 (3.76–7.52)	4.90 (3.02–8.52)	4.86 (3.75–6.8)	4.97 (3.11–6.86)	1.000	1.000	1.000	1.000	1.000	1.000
Serum HDL cholesterol (mmol/L)	1.60 (0.93–2.55)	1.46 (0.92–2.61)	1.56 (0.97–2.19)	1.41 (1.07–2.49)	1.54 (0.83–2.71)	1.61 (1.02–2.18)	1.58 (1.01–2.27)	0.292	1.000	**0.042**	1.000	1.000	1.000
Serum LDL cholesterol (mmol/L)	3.07 (1.87–5.1)	3.16 (1.68–4.99)	3.19 (1.97–5.55)	3.23 (2.34–5.44)	3.11 (1.54–6.25)	2.98 (1.93–4.22)	3.16 (1.71–4.68)	1.000	1.000	1.000	1.000	1.000	1.000
Serum triglycerides (mmol/L)	0.74 (0.35–2.66)	0.85 (0.39–4.69)	0.80 (0.41–2.80)	0.87 (0.39–2.13)	0.76 (0.36–4.76)	0.71 (0.43–2.55)	0.74 (0.42–2.81)	0.516	1.000	0.532	1.000	1.000	1.000
Serum Lp(a) (nmol/L)	1.0 (0.2–105.5)	1.8 (0.2–16.6)	1.15 (0.2–65)	1.4 (0.2–14.2)	1.2 (0.2–12.9)	0.8 (0.4–4.0)	1.6 (0.2–13.5)	1.000	1.000	1.000	1.000	1.000	1.000
Serum CRP (mg/L)	10.4 (5.7–18.85)	10.8 (6.1–29.1)	11.1 (7.7–21.2)	10.6 (6.96–20.2)	10.8 (5.16–16.22)	9.03 (4.89–15.29)	9.56 (4.89–21.39)	0.364	1.000	1.000	1.000	1.000	1.000
Plasma homocysteine (μmol/L)	19.0 (2.39–248.4)	16.8 (2.39–443.72)	16.55 (2.4–307.2)	13.7 (2.4–226.5)	20.64 (2.39–493.16)	16.08 (4.0–264.94)	18.96 (2.39–182.62)	1.000	1.000	1.000	1.000	**0.043**	1.000
Serum uric acid (μmol/L)	246 (143–376)	272 (133–414)	252 (158–431)	279 (163–470)	265.5 (149–477)	254 (170–330)	248 (145–396)	0.063	1.000	**0.046**	0.565	1.000	1.000
Smoking													
Never smoker	54 (60.67%)	76 (61.79%)	26 (68.42%)	32 (68.08%)	95 (84.82%)	24 (70.59%)	21 (53.85%)	0.153	0.601	0.634	**<0.001**	0.515	0.732
Past smoker	21 (23.60%)	30 (24.39%)	6 (15.79%)	10 (21.28%)	8 (7.14%)	7 (20.59%)	10 (25.64%)						
Current smoker	14 (15.73%)	17 (13.82%)	6 (15.79%)	5 (10.64%)	9 (8.04%)	3 (8.82%)	8 (20.51%)						
Contraceptive use													
Never	37 (41.57%)	32 (26.02%)	8 (21.05%)	15 (31.91%)	7 (6.25%)	4 (11.76%)	2 (5.13%)	**0.045**	**0.060**	0.141	**<0.001**	**<0.001**	**<0.001**
In the past	30 (33.71%)	58 (47.15%)	20 (52.63%)	24 (51.06%)	96 (85.71%)	24 (70.59%)	33 (84.61%)						
Current	22 (24.72%)	33 (26.83%)	10 (26.32%)	8 (17.02%)	9 (8.04%)	6 (17.65%)	4 (10.26%)						
Trombophilic gene mutations	1 (1.12%)	11 (8.94%)	5 (13.16%)	4 (8.51%)	8 (7.14%)	4 (11.76%)	4 (10.26%)	**0.015**	**0.003**	**0.029**	**0.040**	**0.007**	**0.014**
**During Gestation**													
Maternal age at delivery (years)	32 (25–43)	32 (21–44)	32 (21–41)	33 (27–51)	34 (26–45)	32 (20–39)	32 (22–42)	1.000	1.000	1.000	1.000	1.000	1.000
GA at delivery (weeks)	39.86 (37.71–41.86)	36.43 (26–41.72)	35.50 (28–41)	39 (35–41.57)	39.64 (37–41.29)	31.15 (24–36.43)	34.0 (24.71–36.86)	**<0.001**	**<0.001**	**0.014**	1.000	**<0.001**	**<0.001**
Mode of delivery													
Vaginal	82 (92.13%)	20 (16.26%)	7 (18.42%)	21 (44.68%)	71 (63.39%)	24 (70.59%)	18 (46.15%)	**<0.001**	**<0.001**	**<0.001**	**<0.001**	**<0.001**	**<0.001**
CS	7 (7.87%)	103 (83.74%)	31 (81.58%)	26 (55.32%)	41 (36.61%)	10 (38.24%)	21 (53.85%)						
Fetal birth weight (g)	3410 (2530–4450)	2430 (610–4490)	1765 (650–3010)	3170 (2275–4440)	3450 (2660–4400)	1575 (542–2820)	2100 (600–2710)	**<0.001**	**<0.001**	1.000	1.000	**<0.001**	**<0.001**
Fetal sex													
Boy	47 (52.81%)	54 (43.90%)	21 (55.26%)	24 (51.06%)	67 (59.82%)	21 (61.76%)	17 (43.59%)	0.200	0.800	0.846	0.319	0.372	0.337
Girl	42 (47.19%)	69 (56.10%)	17 (44.74%)	23 (48.94%)	45 (40.18%)	13 (38.24%)	22 (56.41%)						
Primiparity at index pregnancy													
Yes	43 (48.31%)	99 (80.49%)	37 (97.37%)	31 (65.96%)	50 (44.64%)	23 (67.65%)	29 (74.36%)	**<0.001**	**<0.001**	**0.049**	0.604	0.054	**0.006**
No	46 (51.69%)	24 (19.51%)	1 (2.63%)	16 (34.04%)	62 (55.36%)	11 (32.35%)	10 (25.64%)						
Birth order of index pregnancy													
1st	35 (39.32%)	82 (66.67%)	30 (78.95%)	25 (53.19%)	38 (33.93%)	14 (41.18%)	18 (46.15%)	**<0.001**	**<0.001**	0.191	0.170	0.343	0.792
2nd	33 (37.08%)	20 (16.26%)	3 (7.89%)	9 (19.15%)	44 (39.29%)	9 (26.47%)	13 (33.34%)						
3rd	16 (17.98%)	14 (11.38%)	3 (7.89%)	10 (21.28%)	14 (12.50%)	6 (17.65%)	5 (12.82%)						
4th+	5 (5.62%)	7 (5.69%)	2 (5.26%)	3 (6.38%)	16 (14.28%)	5 (14.70%)	3 (7.69%)						
Total number of pregnancies per patient													
1	8 (8.99%)	38 (30.89%)	12 (31.58%)	10 (21.28%)	10 (8.93%)	2 (5.88%)	9 (23.08%)	**<0.001**	**0.004**	0.121	0.940	0.738	0.085
2	45 (50.56%)	52 (42.28%)	17 (44.74%)	19 (40.42%)	54 (48.21%)	16 (47.06%)	15 (38.46%)						
3+	36 (40.45%)	33 (26.83%)	9 (23.68%)	18 (38.30%)	48 (42.86%)	16 (47.06%)	15 (38.46%)						
Final parity per patient													
1	13 (14.61%)	40 (32.52%)	15 (38.47%)	16 (34.04%)	15 (13.39%)	7 (20.59%)	15 (38.46%)	**0.011**	**0.005**	**0.030**	0.767	0.297	**0.006**
2	62 (69.66%)	66 (53.66%)	21 (55.26%)	26 (55.32%)	75 (66.96%)	25 (73.53%)	22 (56.41%)						
3+	14 (15.73%)	17 (13.82%)	2 (5.26%)	5 (10.64%)	22 (19.64%)	2 (5.88%)	2 (5.13%)						
Infertility treatment													
Yes	4 (4.49%)	25 (20.33%)	8 (21.05%)	6 (12.77%)	17 (15.18%)	4 (11.76%)	6 (15.38%)	**<0.001**	**0.003**	0.079	**0.014**	0.144	**0.034**
No	85 (95.51%)	98 (79.67%)	30 (78.95%)	41 (87.23%)	95 (84.82%)	30 (88.24%)	33 (84.62%)						

Data are presented as median (range) for continuous variables and as number (percent) for categorical variables. Statistically significant results are marked in bold. Continuous variables were compared using Kruskal-Wallis test. Categorical variables were compared using a chi-square test. *p*
^1^ (*p*-value ^1^): the comparison among normal pregnancies and PE; *p*
^2^ (*p*-value ^2^): the comparison among normal pregnancies and FGR; *p*
^3^ (*p*-value ^3^): the comparison among normal pregnancies and GH; *p*
^4^ (*p*-value ^4^): the comparison among normal pregnancies and GDM; *p*
^5^ (*p*-value ^5^): the comparison among normal pregnancies and PTB; *p*
^6^ (*p*-value ^6^): the comparison among normal pregnancies and PPROM. NP, normal pregnancies; PE, preeclampsia; FGR, fetal growth restriction; GH, gestational hypertension; GDM, gestational diabetes mellitus; PTB, preterm birth; PPROM, preterm prelabor rupture of membranes; BMI, body mass index; BP, blood pressure; HDL, high density lipoprotein; LDL, low density lipoprotein; Lp(a), lipoprotein a; CRP, C-reactive protein; CS, Caesarean section; GA, gestational age.

## Data Availability

The data presented in this study are available on request from the corresponding author. The data are not publicly available due to rights reserved by founders.

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
