# Peer review of "Association Analysis in Young and Middle-Aged Mothers—Relation between Expression of Cardiovascular Disease Associated MicroRNAs and Abnormal Clinical Findings"

_jpm, 2021, doi:10.3390/jpm11010039_

Round 1

Reviewer 1 Report

  • This reviewer thinks that daily exercise may influence both obesity, hypertension, and miRNA. Can you show data regarding daily exercise of the study patients?

Author Response

Reviewer 1

This reviewer thinks that daily exercise may influence both obesity, hypertension, and miRNA. Can you show data regarding daily exercise of the study patients? 

Reviewer 1 answers

As requested, these data were added into the manuscript.

  1. Results

2.9. The effect of physical activity on postpartal microRNA expression profile, BMI values, SBP and DBP values

Patients were divided into two groups, those ones who were not used to run any sport activity (n= 295) and those ones who were running some sport activities (n= 183). Minimum women were running sport activities on daily basis or several times per week. Most women, who did any sport activity, made it mainly from time to time for leisure.

The association analysis between physical activity and microRNA gene expression showed decreased levels of microRNA gene expression in women used to run any sport activity from time to time, however any analysis did not achieve statistical significance (Table S2).

The association analysis between physical activity and BMI values also showed decreased levels of BMI values in women used to run any sport activity from time to time, however the analysis have not yet achieved the statistical significance (Mann-Whitney test, p= 0.186) (Figure S11). Similarly, the impact of physical activity on BMI values (below 25 vs. above 25) has not yet achieved statistical significance, when logistic regression was used (OR 0.774, 95% CI: 0.5280 to 1.1359, p= 0.1892).

On the other hand, physical activity mentioned above had no impact on SBP values (Mann-Whitney test, p= 0.852) (Figure S12) in our cohort of patients. Similarly, physical activity mentioned above had also no impact on DBP values (Mann-Whitney test, p= 0.735) (Figure S13). Likewise, no effect of physical activity mentioned above was observed in women with the presence of systolic hypertension (above 140 mmHg vs. below 140 mmHg) (OR 0.8908, 95% CI: 0.4019 to 1.9744, p= 0.775) and diastolic hypertension (above 90 mmHg vs. below 90 mmHg) (OR 0.8527, 95% CI: 0.4425 to 1.6432, p= 0.632).

Table S2. The association analysis between occasional physical activity and microRNA gene expression

MicroRNA

Median

Women without sport activity

Median

Women with sport activity

Mann-Whitney test

P value

miR-1-3p

0.116

0.084

0.322

miR-16-5p

1.093

1.053

0.254

miR-17-5p

1.316

1.229

0.325

miR-20a-5p

1.305

1.164

0.391

miR-20b-5p

1.332

1.236

0.447

miR-21-5p

0.268

0.257

0.439

miR-23a-3p

0.175

0.138

0.220

miR-24-3p

0.244

0.214

0.584

miR-26a-5p

0.500

0.440

0.213

miR-29a-3p

0.283

0.264

0.383

miR-92a-3p

1.809

1.824

0.770

miR-100-5p

0.0015

0.0014

0.683

miR-103a-3p

1.282

1.177

0.253

miR-125b-5p

0.0036

0.0031

0.367

miR-126-3p

0.227

0.210

0.620

miR-130b-3p

0.480

0.456

0.359

miR-133a-3p

0.107

0.089

0.235

miR-143-3p

0.025

0.020

0.231

miR-145-5p

0.094

0.089

0.531

miR-146a-5p

1.024

0.939

0.442

miR-155-5p

0.963

0.921

0.815

miR-181a-5p

0.216

0.202

0.193

miR-195-5p

0.079

0.067

0.875

miR-199a-5p

0.046

0.040

0.370

miR-210-3p

0.092

0.091

0.886

miR-221-3p

0.514

0.458

0.219

miR-342-3p

2.500

2.485

0.873

miR-499a-5p

0.212

0.164

0.356

miR-574-3p

0.137

0.125

0.331

Figure S11. The association analysis between physical activity and BMI values

Figure S12. The association analysis between physical activity and SBP values

Figure S13. The association analysis between physical activity and DBP values

Reviewer 2 Report

This is a very interesting paper about the postpartum alterations in gene expression in microRNAs in association with diabetes mellitus, cardiovascular and cerebral disease after pregnancy, induced by pregnancy-related complications such as preeclampsia, IUGR, gestational diabetes mellitus, gestational hypertension, preterm prelabor rupture of membranes or spontaneous preterm birth. Nevertheless, I have several questions and comments for the authors:

  1. The title is too long and it is not clear enough to understand the meaning of the study.
  2. In the results part, firstly the authors analyze the influence of pregnancy-related complications and different microRNA gene expressions. It would be much more interesting from my perspective to split these pregnancy complications into the different diseases to know exactly which microRNA gen are related to. Which ones are related to PE or diabetes? How does the rupture of membranes influences these gene expressions? Are the same for PE or spontaneous preterm birth? This is a very important issue. Also, how many cases are there of each pregnancy complications cases included in the study?
  3. Results of 2.2. The authors conclude no association between different microRNA gene expression and different maternal characteristics or treatments such as hormone contraceptive use. This is should be better explained and more specific. 
  4. Ultimately, the results and the manuscript itself should be presented in a more clear way and shorter.

Author Response

Reviewer 2

This is a very interesting paper about the postpartum alterations in gene expression in microRNAs in association with diabetes mellitus, cardiovascular and cerebral disease after pregnancy, induced by pregnancy-related complications such as preeclampsia, IUGR, gestational diabetes mellitus, gestational hypertension, preterm prelabor rupture of membranes or spontaneous preterm birth. Nevertheless, I have several questions and comments for the authors:

 Point 1

The title is too long and it is not clear enough to understand the meaning of the study.

Answer to point 1

As suggested, the title of the manuscript was revised and significantly shortened.

Association analysis in young and middle-aged mothers – relation between expression of cardiovascular disease associated microRNAs and abnormal clinical findings

Point 2

In the results part, firstly the authors analyze the influence of pregnancy-related complications and different microRNA gene expressions. It would be much more interesting from my perspective to split these pregnancy complications into the different diseases to know exactly which microRNA gen are related to. Which ones are related to PE or diabetes? How does the rupture of membranes influences these gene expressions? Are the same for PE or spontaneous preterm birth? This is a very important issue. Also, how many cases are there of each pregnancy complications cases included in the study?

Answer to point 2

The purpose of this study was to perform association analysis between microRNA gene expression and abnormal clinical findings. Individual postpartal microRNA expression profiles have already been published for women after pregnancies affected with gestational hypertension, preeclampsia, fetal growth restriction, and gestational diabetes mellitus. This information was mentioned in original manuscript in introduction section, please see the information provided below. The microRNA expression profile in patients with a history of preterm birth and PPROM will be assessed as well, however, later as a subject of another study, when sufficient amount of patients will be collected. The number of cases involved in individual categories were also stated in original manuscript in section “Materials and Methods” and also in “Table 1 Characteristics of cases and controls”. Please see the information provided below.

  1. Introduction

Recently, we have shown that a proportion of mothers with a history of pregnancy-related complications such as gestational hypertension (GH), preeclampsia (PE), fetal growth restriction (FGR), and gestational diabetes mellitus (GDM) had alterations in microRNA expression profiles in whole peripheral blood (leukocytes) that may contribute besides other factors to the onset of diabetes mellitus, cardiovascular and cerebrovascular diseases [20, 21]. 

A previous occurrence of GH, PE, FGR, and GDM was demonstrated to be associated in a proportion of mothers with alterations in gene expression of miR-1-3p, miR-17-5p, miR-20a-5p, miR-20b-5p, miR-29a-3p, miR-100-5p, miR-125b-5p, miR-126-3p, miR-130b-3p, miR-133a-3p, miR-143-3p, miR-145-5p, miR-146a-5p, miR-181a-5p, miR-199a-5p, miR-221-3p, and miR-499a-5p [20, 21].

Furthermore, a history of GDM was shown to be associated with dysregulation of miR-16-5p, miR-21-5p, miR-23a-3p, miR-24-3p, miR-26a-5p, miR-103a-3p, miR-195-5p, miR-342-3p, and miR-574-3p in a substantial proportion of mothers [21].

  1. Materials and Methods

4.1. Participants

The prospective study running from 8/2016 - 10/2020 included Caucasian women after normal pregnancies (n= 89) and pregnancies complicated with gestational diabetes mellitus (n= 112), gestational hypertension (n= 47), preeclampsia (n= 123), fetal growth restriction (n= 38), preterm prelabor rupture of membranes (n= 39), and spontaneous preterm birth (n= 34). The clinical data of women are displayed in Table 1.

Point 3

Results of 2.2. The authors conclude no association between different microRNA gene expression and different maternal characteristics or treatments such as hormone contraceptive use. This is should be better explained and more specific. 

Answer to point 3

As requested, this part was revised and expanded. Please see below.

2.2. No association between postpartal expression of diabetes/cardiovascular/cerebrovascular disease associated microRNAs and actual hormonal contraceptive use, active smoking of cigarettes, total serum cholesterol levels, serum HDL cholesterol levels, serum LDL cholesterol levels, serum triglycerides levels, serum lipoprotein A levels, serum CRP levels, serum uric acid levels, and plasma homocysteine levels

No association between expression of diabetes/cardiovascular/cerebrovascular disease associated microRNAs and actual hormonal contraceptive use, and active smoking of cigarettes was observed in young and middle-aged mothers. In detail, there was no difference in microRNA gene expression between never users and current users of any hormonal contraceptives (combined oral contraceptives, the progestogen-only pills, oral selective estrogen receptor modulators, intrauterine device, contraceptive ring, or the contraceptive implant). Furthermore, there was no difference in microRNA gene expression between never smokers and current smokers of tabacco cigarettes.

In addition, no association between microRNA expression levels and serum levels of total cholesterol, HDL cholesterol, LDL cholesterol, triglycerides, lipoprotein A, CRP, and uric acid was found. MicroRNA gene expression did not also differ in relation to plasma homocysteine levels. In detail, there was no difference in microRNA gene expression between mothers with abnormal and normal serum or plasma levels of these biochemical analytes. 

Point 4

Ultimately, the results and the manuscript itself should be presented in a more clear way and shorter.

Answer to point 4

The authors paid substantial effort to careful presentation of the data and preparation of the whole manuscript.  Huge amount of data is presented in the manuscript. We regret, that the reviewer has a different opinion concerning the data presentation in our manuscript. Since the reviewer did not make any concrete suggestion how the data should be better organized and how the presentation of the data could be improved, we decided to leave the data presentation as it was in our original manuscript. Thank you for your understanding.

Round 2

Reviewer 2 Report

The authors made most of the suggested changes.